# Dynamic Expression of EpCAM in Primary and Metastatic Lung Cancer Is Controlled by Both Genetic and Epigenetic Mechanisms

**DOI:** 10.3390/cancers14174121

**Published:** 2022-08-25

**Authors:** Yeting Cui, Jiapeng Li, Xiyu Liu, Lixing Gu, Mengqing Lyu, Jingjiao Zhou, Xiaoyu Zhang, Yu Liu, Haichuan Zhu, Tongcun Zhang, Fan Sun

**Affiliations:** 1College of Life Sciences and Health, Wuhan University of Science and Technology, Wuhan 430065, China; 2Institute of Biology and Medicine, Wuhan University of Science and Technology, Wuhan 430065, China; 3College of Science, Wuhan University of Science and Technology, Wuhan 430065, China

**Keywords:** EpCAM, lung cancer, primary, metastasis, genetic, epigenetic, macrophage

## Abstract

**Simple Summary:**

Epithelial cell adhesion molecule (EpCAM) is a tumor marker widely used in both basic studies and clinics. However, our study demonstrates that EpCAM expression is strongly upregulated by gene amplification and promoter hypomethylation in primary lung tumors, but severely downregulated by epigenetic repression (including promoter hypermethylation and histone deacetylation), tumor-associated macrophages (TAMs), and TAMs-derived TGFβ in metastatic lung tumors. DNMT inhibitor 5-aza-dC, HDAC inhibitor MS-275, and TGFβ neutralizing antibody are able to restore EpCAM expression in highly metastatic lung cancer cells. These findings disclose that multiple mechanisms contribute to the dynamic expression patterns of EpCAM in primary and metastatic lung tumors, redefining the application of EpCAM as a biomarker in tumor cell identification and isolation in specific cancers and clinical stages.

**Abstract:**

Although great progress has been achieved in cancer treatment in the past decades, lung cancer remains the leading cause of cancer death, which is partially caused by the fact that most lung cancers are diagnosed at advanced stages. To improve the sensitivity and specificity of lung cancer diagnosis, the underlying mechanisms of current diagnosis methods are in urgent need to be explored. Herein, we find that the expression of EpCAM, the widely used molecular marker for tumor cell characterization and isolation, is strongly upregulated in primary lung tumors, which is caused by both gene amplification and promoter hypomethylation. In contrast, EpCAM expression is severely repressed in metastatic lung tumors, which can be reversed by epigenetic drugs, DNMT inhibitor 5-aza-dC and HDAC inhibitor MS-275. Moreover, tumor-associated macrophages (TAMs) impede EpCAM expression probably through TGFβ-induced EMT signaling. These findings unveil the dynamic expression patterns of EpCAM and differential roles of epigenetic modification in EpCAM expression in primary and metastatic lung tumors, providing novel insights into tumor cell isolation and lung cancer diagnosis.

## 1. Introduction

Lung cancer is the leading cause of cancer death with the highest incidence rate (13%) and mortality rate (21%) all over the world [1]. In the past decades, great progress has been made in lung cancer treatment including targeted therapy, immunotherapy, etc.; however, the 5-year survival rate in patients with lung cancers is still as low as 22% [2,3,4]. One of the major reasons is that most lung cancers are diagnosed at advanced stages because of insensitivity and low specificity of current diagnosis methods, including low-dose computed tomography (LDCT), sputum cytology, and tumor biomarker detection [5,6]. Therefore, to determine the underlying mechanisms of these technologies is urgently needed to improve the sensitivity and specificity of diagnosis in patients with lung cancers.

EpCAM, epithelial cell adhesion molecule, is a highly conserved membrane protein in vertebrates [7,8]. Physiologically, EpCAM is ubiquitously expressed on epithelial cells in organs including lung, colon, bladder, and breast, whereas in pathological conditions, EpCAM is specifically expressed in cancers originated from epithelial cells [7,8]. Moreover, EpCAM expression is frequently upregulated in a variety of human cancers, including lung cancer, breast cancer, cervical cancer, and pancreatic cancer [9]. Given the expression characteristics, EpCAM is utilized as a standard molecular marker for identification and isolation of cancer cells from blood, bone marrow, lymph nodes, and biopsy samples [10,11]. With its widely application in basic and translational research, however, minimal or no expression of EpCAM is found on circulating tumor cells (CTCs) in some cancer patients [12,13]. So far, it is unclear that how EpCAM expression is strong upregulated in tumor cells but sometimes diminishes.

Multiple mechanisms are involved in the regulation of gene expression, including genetic, epigenetic, and transcription factor [14,15]. Gene amplification leads to upregulation of oncogene, such as EGFR and C-MYC, while loss of heterozygosity (LOH) and homozygous deletion (HD) impair the expression of tumor suppressor gene, such as TP53 and Rb [16,17,18]. Promoter hypermethylation and histone deacetylation synergistically contribute to the repression of PDLIM2 and E-cadherin in many types of human cancers [19,20,21]. In most cases, transcription factors are the major regulators of gene expression in both physiological and pathological conditions, e.g., NF-κB and STAT3 regulation of survival and proliferative genes [22,23].

In the present study, combining public databases and our experiments, the expression of EpCAM was extensively examined in both primary and metastatic lung tumors. The underlying mechanisms accounting for dynamic expression of EpCAM in different stages of lung cancer were explored with respect to both genetic and epigenetic regulation manners. In addition, the effect of tumor microenvironment on tumor cell EpCAM expression was investigated. These findings about EpCAM expression regulation will provide novel insights into tumor cell isolation and lung cancer diagnosis.

## 2. Materials and Methods

### 2.1. Public Data Mining

The gene profiles about expression, mutation, methylation, and copy number of *epcam* in human cancers were retrieved from The Cancer Genome Atlas (TCGA) database (http://xena.ucsc.edu) accessed on 20 January 2021. Other data about EpCAM expression in mouse lung tumors and cell lines were downloaded from NCBI GEO database (https://www.ncbi.nlm.nih.gov/gds) accessed on 25 January 2021 as described [19,24]. All information in detail are indicated in Figure legends.

### 2.2. Mouse and Tumor Cells

FVB/N, BALB/c, and C57BL/6 mice originally from Beijing HFK Bioscience Co., Ltd. were housed in pathogen-free conditions and used according to protocols approved by the Animal Ethics Committee of Wuhan University of Science and Technology. To induce lung tumorigenesis, 1 g/kg body weight of urethane was intraperitoneally injected once a week for six consecutive weeks in 6-week-old female FVB/N wildtype mice. After 6-week tumor initiation and 6-week tumor progression, the mice were sacrificed for lung tumor analysis. For spontaneous primary lung tumor and metastatic lung tumor, female FVB/N wildtype mice were maintained for 24 months and sacrificed for tumor examination. Mouse lung tumor cells (LAP0297, MAD109, and LLC) and human lung tumor cells (Kras mutant: A549, Calu-6, H727, and H460; EGFR mutant: H1650, H1975, H3255, and HCC827) were cultured in RPMI 1640 medium supplemented with 10% FBS and 1% penicillin/streptomycin at 37 °C in humidified 5% CO_2_ incubator as described [24,25].

### 2.3. Immunoblotting Analysis

An equal amount of protein extracts from the indicated normal lung tissues and lung tumors were separated on 10% SDS-PAGE gels and transferred onto nitrocellulose membrane. The membrane was blocked with 5% non-fat milk in freshly prepared TBST buffer (25 mM Tris-HCl pH 7.5, 150 mM NaCl, and 0.1% Tween 20), then sequentially incubated primary antibodies and secondary antibodies. At last, the signal was detected by ECL system as described [26]. All antibodies used in this study are shown in Appendix A.

### 2.4. FACS Analysis

The single cell suspensions were blocked with αCD16/CD32 (mouse) or TruStain FcX (human), then stained with the antibody against cell surface antigen. CD45 was utilized as a marker to distinguish immune cells from other cells in primary tumor cell suspensions and in vitro cell cocultures. Data were acquired by Accuri C6 or LSRFortessa I and analyzed by FlowJo software as described [27]. All antibodies used in this study are shown in Appendix A.

### 2.5. RT-PCR Analysis

Total RNAs were extracted from the indicated normal lung tissues, lung tumors, and lung cancer cells, and then subjected to RNA reverse transcription and real-time PCR as described [19,22,24]. All primers used for qPCR in this study are shown in Appendix A.

### 2.6. Statistical Analysis

Two-tailed, unpaired Student’s *t* test was used to assess significance of differences between two groups. All data are represented as bars (means ± SEM) with sample dots. The sample numbers for RT-PCR, FACS, immunoblotting, and sequencing are indicated in the Figures or Figure legends. The *p* values are indicated as * *p* < 0.05, ** *p* < 0.01, ns, not statistically significant.

## 3. Results

### 3.1. EpCAM Expression Is Upregulated in Primary Lung Cancer

EpCAM is widely used as a selection marker for tumor cell identification and isolation in both human and mouse studies. In line with its application, TCGA and CPTAC data revealed that EpCAM expression was strongly upregulated at both RNA and protein levels in human primary lung cancer compared to normal lung tissues, respectively (Figure 1A,B), which was supported by the qPCR data showing increased EpCAM mRNA level in primary lung tumors compared to their matched normal lung tissues (Figure 1C). To verify these findings in human primary lung cancer, GSE31013 data were analyzed and found EpCAM upregulation in murine primary lung tumors spontaneously derived from B6C3F1 mice (Figure 1D), which was further validated by the elevation of EpCAM mRNA levels in both spontaneous and carcinogen urethane-induced primary lung tumors from FVB/N mice (Figure 1E,F). In addition, enhanced protein expression of EpCAM was confirmed in murine primary lung tumors in comparison with normal lung tissues by FACS and immunoblotting assays (Figure 1G–I and Appendix A) These data together indicated that EpCAM is upregulated in primary lung cancer.

### 3.2. Gene Amplification Induces EpCAM Expression in Primary Lung Cancer

How do primary lung cancers express high levels of EpCAM? Initially, the copy number of *epcam* gene was examined in TCGA LUNG cohort. Overall, a significant increase of copy number was observed in *epcam* gene loci (Figure 2A), while as many as 41% of lung cancers harbored low-to-high *epcam* gene amplification (Figure 2B). Correlation between copy number and expression level exhibited that about 64% of patients with primary lung cancers experienced both *epcam* gene amplification and increased expression (Figure 2C). Moreover, cigarette smoking, the major cause of lung cancer, significantly promoted *epcam* gene amplification (Figure 2D). These data suggest that gene amplification is one of the mechanisms to upregulate EpCAM expression in primary lung cancer.

### 3.3. Promoter Hypomethylation Induces EpCAM Expression in Primary Lung Cancer

About 17% of lung cancer patients possessed high EpCAM expression levels but no gene amplification (Figure 2C), suggesting additional mechanism(s) involved in upregulation of EpCAM expression. Besides genetic modification, epigenetic regulation frequently controlled the activation and repression of genes in cancers [14,15]. Herein, DNA methylation data from TCGA LUNG cohort showed that compared with normal lungs, the methylation level in *epcam* promoter was significantly decreased in primary lung tumors (Figure 3A), with the biggest difference of methylation level at CpG site corresponding to probe cg03706175 (Figure 3B). About 70% of patients with primary lung cancers harbored both *epcam* promoter hypomethylation and increased expression (Figure 3C). Furthermore, cigarette smoking, the major cause of lung cancer, severely induced hypomethylation in *epcam* promoter (Figure 3D). These data suggest that promoter hypomethylation is one of the mechanisms to induce EpCAM expression in primary lung cancer.

### 3.4. EpCAM Expression Is Downregulated in Metastatic Lung Cancer

Previous data showed upregulation of EpCAM in primary lung cancer (Figure 1), leading to a question of EpCAM expression in metastatic lung cancer. To address this issue, public NCBI GEO database was employed in the present study. In human lung cancer CL1 cells, EpCAM expression was severely impaired in highly invasive cells in comparison with poorly invasive ones (Figure 4A). The EMT status induced by TGFβ strongly impeded EpCAM expression in A549, HCC827, and H358 lung cancer cells (Figure 4B). These data of EpCAM repression in human lung cancer were also confirmed in murine lung cancer. As shown in Figure 4C, compared with primary lung tumors, EpCAM expression was significantly decreased in metastatic lung tumors from Kras^G12D^p53^flox^ mice. Moreover, metastatic lung tumors expressed minimal EpCAM protein in contrast to high levels of EpCAM protein in urethane-induced and spontaneous primary lung tumors from FVB/N mice (Figure 4D and Appendix A). These data suggest that EpCAM expression is downregulated in metastatic lung tumors.

In addition, low EpCAM protein levels were found in highly metastatic human lung cancer cells, including A549, Calu-6, H727, and H460 (Figure 5A), but high in poorly metastatic lung cancer cells, including H1650, H1975, H3255, and HCC827 (Appendix A). Consistently, minimal or no expression of EpCAM RNA and protein was detected in highly metastatic murine lung cancer cells, including LLC, MAD109, and LAP0297 (Figure 5B–D). Taken together, these data suggest that, similar to metastatic lung tumors, EpCAM expression is also repressed in highly metastatic lung cancer cells.

### 3.5. Epigenetic Drugs Restore EpCAM Expression in Highly Metastatic Lung Cancer Cells

How is EpCAM expression in lung cancer switched from upregulation in primary tumor to downregulation in metastatic tumor? Promoter methylation was one of the mechanisms accounting for EpCAM repression in highly metastatic lung cancer cell A549, which was evidenced by strong restoration of EpCAM expression by DNMT inhibitor 5-aza-dC in both time- and dose-dependent manners (Figure 6A). Histone deacetylation, another epigenetic regulation, also caused EpCAM repression, which could be reversed by HDAC inhibitor MS-275 in highly metastatic lung cancer cells Calu-6, H727, H460, and A549 (Figure 6B). Given that promoter methylation and histone deacetylation always synergized in regulation of gene expression, the effect of DNMT inhibitor, HDAC inhibitor, and their combination on EpCAM restoration was examined. Indeed, the strongest induction of EpCAM expression was observed by combination treatment in comparison with individual inhibitor in both human and murine highly metastatic cancer cells (Figure 6C,D). These data indicate that epigenetic mechanisms partially contribute to EpCAM repression in metastatic lung tumors.

### 3.6. Macrophage-Derived TGFβ Represses EpCAM Expression in Lung Cancer Cells

Besides epigenetic regulation, EMT signaling, the origin of metastasis, may also contribute to EpCAM downregulation, which was supported by negative association between EMT status and EpCAM expression level in human lung cancers (Appendix A). The prototypical EMT inducer TGFβ and its downstream signaling molecule SNAI2 were inversely correlated with EpCAM expression in human lung cancers (Figure 7A and Appendix A). TGFβ overexpression strongly inhibited the expression of EpCAM in both human and murine lung cancer cells (Figure 7B–D), but stimulated the expression of SNAI2, which could bind to the E-box in *epcam* promoter to suppress its transcription (Appendix A). These data suggest that TGFβ-induced EMT signaling causes the downregulation of EpCAM in lung cancer.

Given that tumor microenvironment, in particular tumor-associated macrophage, is involved in EMT and metastasis in lung cancer [28,29], the effect of macrophage on tumor cell EpCAM expression was examined. Indeed, all macrophages, including primary (AM, PEC), cell line (RAW 264.7), and BMM, were able to suppress EpCAM expression on lung tumor cells (Figure 7E), which could be reversed by TGFβ neutralizing antibody (Figure 7F). Taken together, these data suggest that macrophages in tumor microenvironment likely produce TGFβ to repress EpCAM expression in lung cancer cells.

## 4. Discussion

Precise detection and characterization of tumor cells are critical for cancer diagnosis, treatment, and patient survival, particularly for lung cancer [5,6,10,11,12,13]. The current biomarkers used for tumor cell isolation, e.g., EpCAM, are not effective due to their insensitivity and low specificity, which leads to the requirement of mechanistic studies on expression regulation of these tumor biomarkers. In the present study, EpCAM expression is found strongly upregulated in primary lung tumors, which is caused by both gene amplification and promoter hypomethylation, but severely repressed in metastatic lung tumors, which can be reversed by epigenetic drugs, DNMT inhibitor 5-aza-dC, and HDAC inhibitor MS-275. Additionally, tumor-associated macrophage (TAM), the metastasis stimulator, is capable of secreting TGFβ to impair EpCAM expression probably through SNAI2. These findings disclose that multiple mechanisms including genetic, epigenetic, and transcription factors contribute to dynamic expression of EpCAM in primary and metastatic lung tumors, providing novel insights into tumor cell isolation and lung cancer diagnosis.

EpCAM is a widely used biomarker for tumor cell isolation in both basic studies and clinics, because of its strong upregulation in a variety of human cancers, including lung cancer, breast cancer, cervical cancer, pancreatic cancer, and stomach cancer (Appendix A). However, the application of EpCAM as a biomarker for tumor cell isolation from blood, bone marrow, lymph nodes, and biopsy samples is not suitable for colon cancer, kidney cancer, liver cancer, and thyroid cancer, in which EpCAM exhibits repressed or comparable expression levels compared to normal tissues (Appendix A). These results not only support the concept of EpCAM as a selection tumor marker, but also refine its application in tumor cell identification and isolation in specific cancers.

Gene amplification is one of the regulation manners to induce EpCAM expression in human primary cancers, including lung, bladder, cervical, and esophagus (Appendix A). However, no significant gene amplification of *epcam* was observed in BRCA, CHOL, and STAD, which exhibit EpCAM upregulation (Appendix A). These results suggest that multiple mechanisms together, but not gene amplification only, contribute to EpCAM upregulation in human primary tumors. Indeed, promoter hypomethylation is one of these mechanisms (Figure 3).

Mutation of *epcam* gene is linked to lynch syndrome and congenital tufting enteropathy [30] but is rare (0.7%) and has neglectable effect on its expression in primary lung cancers (Appendix A). However, both genetic (gene amplification) and epigenetic (promoter hypomethylation) modification contribute to EpCAM upregulation in primary lung cancers (Figure 2 and Figure 3). Moreover, a minority of patients with lung cancer (6.67%) harbors gene loss and promoter hypermethylation, but EpCAM upregulation, which is likely caused by smoking-induced ROS (Appendix A). These findings suggest that EpCAM upregulation in primary lung cancer is subtly regulated at various levels from genetic and epigenetic to factors in tumor microenvironment.

In contrast to upregulation in primary lung tumors, EpCAM is severely repressed in metastatic lung tumors, which is mediated by promoter hypermethylation, histone deacetylation, and TGFβ from macrophage in tumor microenvironment (Figure 4, Figure 5, Figure 6 and Figure 7 and Appendix A). Thus, epigenetic regulation is one of the mechanisms controlling EpCAM expression in both primary and metastatic lung cancers and its expression switch from upregulation to downregulation during tumor progression. This phenomenon is rare and not well studied. Further studies will clarify the dynamic profile of proteins bound to *epcam* promoter during lung cancer progression to figure out the mechanism, shifting epigenetic activation in primary tumor to epigenetic repression in metastatic tumor. In addition, macrophage, the major cell type constituting tumor microenvironment, produces TGFβ to modulate EpCAM expression on tumor cells through TGFβ-SNAI2 axis to promote EMT and metastasis in lung cancer. This crosstalk between macrophage and tumor cells is beneficial for understanding of cancer metastasis regulation by tumor microenvironment.

Given the differential expression pattern of EpCAM in primary and metastatic lung tumors, the function of EpCAM in different lung cancer stages may also be different. The protumor activity of EpCAM was adequately reported in primary tumors [7,8]; however, few and contradictory findings about regulation of EGFR-EMT axis by EpCAM (or its extracellular domain EpEX) were found in literature [31,32,33,34,35]. Our preliminary data identified an inhibition effect of EpCAM reconstitution on metastasis in lung cancer. However, the underlying mechanism is not clear yet. Further studies will be carried out to dig out the mechanism of metastasis inhibition by EpCAM overexpression in lung cancer cells.

## 5. Conclusions

In summary, we found EpCAM expression upregulated in primary lung tumors, which was caused by gene amplification and promoter hypomethylation. However, in metastatic lung tumors, EpCAM expression is intensely repressed by macrophage-derived TGFβ, promoter hypermethylation, and histone deacetylation, which could be reversed by TGFβ neutralizing antibody and epigenetic drugs, DNMT inhibitor 5-aza-dC and HDAC inhibitor MS-275, representing novel therapeutic strategies to treat metastatic lung cancer. Whether genetic modification is involved in EpCAM downregulation in metastatic lung tumors and how EpCAM repression affects cancer metastasis are to be addressed in our further studies. Taken together, these findings demonstrated that multiple mechanisms including genetic, epigenetic, and tumor microenvironment modulated the dynamic expression patterns of EpCAM in primary and metastatic lung tumors, shedding light on cell isolation, diagnosis, and treatment in lung cancer.

## Figures and Tables

**Figure 1 cancers-14-04121-f001:**
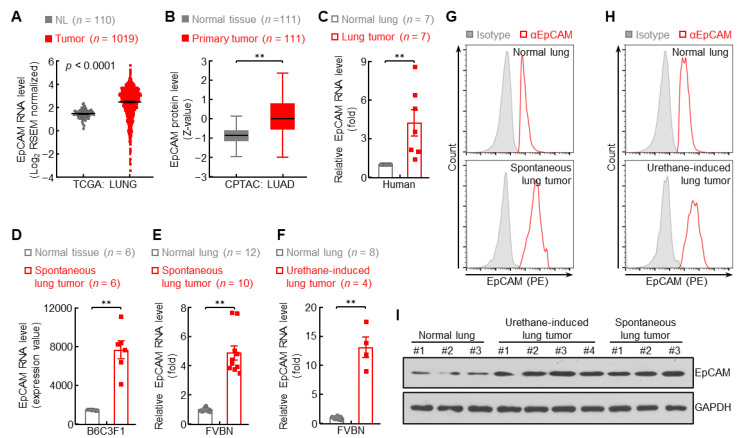
EpCAM expression is upregulated in primary lung cancer. (**A**) TCGA LUNG data showing increased EpCAM RNA expression in human primary lung tumor (Tumor: red dot) compared to normal lung (NL: grey dot). (**B**) CPTAC LUAD data showing increased EpCAM protein expression in human primary lung tumor. (**C**) qPCR data showing increased EpCAM RNA expression in human primary lung tumor compared to normal lung tissue from the same patients. (**D**) GSE31013 data showing increased EpCAM RNA expression in spontaneous primary lung tumor from B6C3F1 mice. (**E**,**F**) qPCR data showing increased EpCAM RNA expression in spontaneous (**E**) and urethane-induced (**F**) primary lung tumor from FVBN mice. (**G**,**H**) FACS data showing increased EpCAM protein expression in spontaneous (**G**) and urethane-induced (**H**) primary lung tumor from FVBN mice (*n* = 4). (**I**) Western blot data showing increased EpCAM protein expression in urethane-induced and spontaneous primary lung tumor from FVBN mice. Date are representative of at least three independent experiments with similar results (**C**,**E**–**I**). Student’s *t* test (two-tailed, unpaired) was performed. Data represent means ± SEM (**A**,**C**–**F**) and means ± SD (**B**). ** *p* < 0.01.

**Figure 2 cancers-14-04121-f002:**
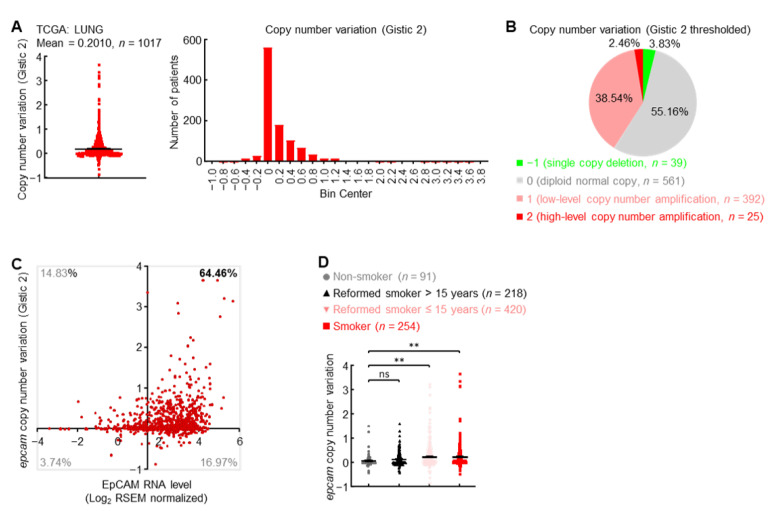
Gene amplification induces EpCAM expression in primary lung cancer. (**A**) TCGA LUNG Gistic 2 data showing *epcam* gene amplification in human primary lung cancer. (**B**) TCGA LUNG Gistic 2 thresholded data showing distribution of *epcam* copy number variation in human primary lung cancer. (**C**) TCGA LUNG data showing *epcam* gene amplification in about 64% of lung cancers with increased EpCAM expression. Axis *x* and *y* cross at mean value (1.45, 0) of EpCAM expression level and copy number variation in normal lung. (**D**) TCGA LUNG data showing positive association between smoking and *epcam* gene amplification. Student’s *t* test (two-tailed, unpaired) was performed. Data represent means ± SEM (**D**). ** *p* < 0.01; ns, not statistically different.

**Figure 3 cancers-14-04121-f003:**
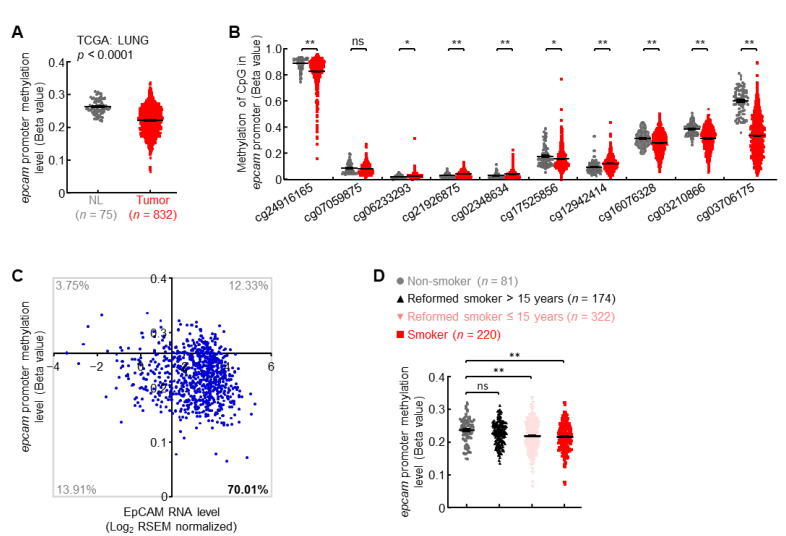
Promoter hypomethylation induces EpCAM expression in primary lung cancer. (**A**) TCGA LUNG data showing decreased *epcam* promoter methylation in human primary lung cancer. (**B**) TCGA LUNG data showing methylation status of each CpG site in *epcam* promoter in human primary lung cancer. (**C**) TCGA LUNG data showing *epcam* promoter hypomethylation in about 70% of lung cancers with increased EpCAM expression. Axis *x* and *y* cross at mean value (1.45, 0.26) of EpCAM expression level and promoter methylation level in normal lung. (**D**) TCGA LUNG data showing negative association between smoking and *epcam* promoter methylation. Student’s *t* test (two-tailed, unpaired) was performed. Data represent means ± SEM (**A**,**B**,**D**). * *p* < 0.05; ** *p* < 0.01; ns, not statistically different.

**Figure 4 cancers-14-04121-f004:**
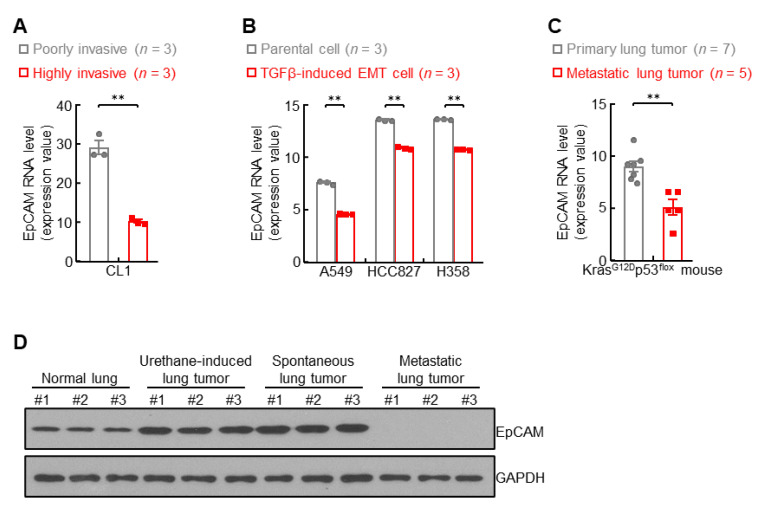
EpCAM expression is downregulated in metastatic lung cancer. (**A**) GSE42407 data showing decreased EpCAM RNA expression in highly invasive human lung cancer cell CL1. (**B**) GSE49644 data showing decreased EpCAM RNA expression in TGFβ-induced EMT cells. (**C**) GDS4402 data showing decreased EpCAM RNA expression in lung cancer cells derived from metastatic lung tumors compared to primary lung tumors from Kras^G12D^p53^flox^ mice. (**D**) Western blot data showing decreased EpCAM protein expression in breast metastatic lung tumor from FVBN mice. Data are representative of three independent experiments with similar results (**D**). Student’s *t* test (two-tailed, unpaired) was performed. Data represent means ± SEM (**A**–**C**). ** *p* < 0.01.

**Figure 5 cancers-14-04121-f005:**
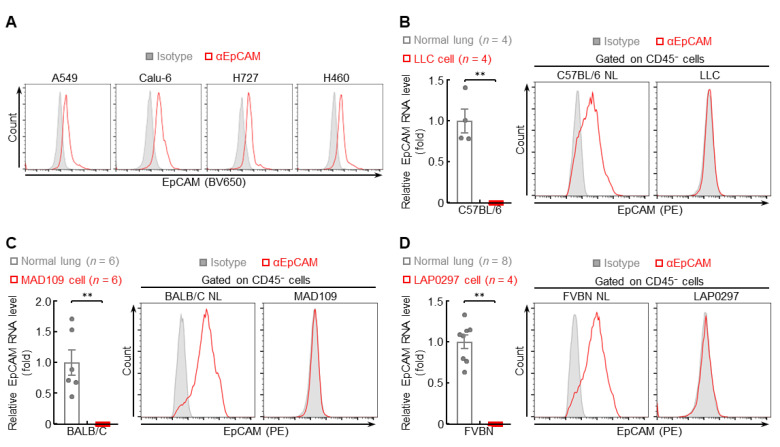
EpCAM expression is repressed in highly metastatic lung cancer cells. (**A**) FACS data showing decreased EpCAM protein expression in human highly metastatic lung cancer cells (*n* = 3). (**B**–**D**) qPCR and FACS data showing repressed EpCAM expression in murine highly metastatic lung cancer cells (FACS: *n* = 4). Data are representative of at least three independent experiments with similar results. Student’s *t* test (two-tailed, unpaired) was performed. Data represent means ± SEM (**B**–**D**). ** *p* < 0.01.

**Figure 6 cancers-14-04121-f006:**
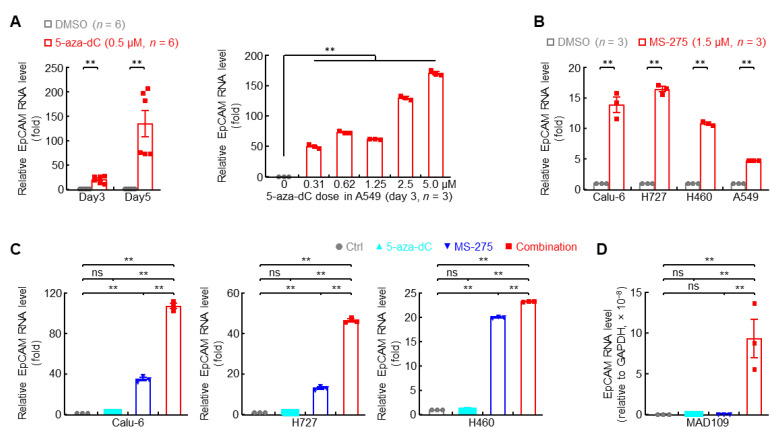
Epigenetic drugs restore EpCAM expression in highly metastatic lung cancer cells. (**A**) qPCR data showing recovered EpCAM RNA expression by DNMT inhibitor (5-aza-dC) in human highly metastatic lung cancer cell A549. (**B**) qPCR data showing restored EpCAM RNA expression by HDAC1 inhibitor (MS-275, 72 hr) in human highly metastatic lung cancer cells. (**C**,**D**) qPCR data showing synergistic induction of EpCAM RNA expression by DNMT inhibitor (5-aza-dC) and HDAC1 inhibitor (MS-275) in human (**C**) and murine (**D**) highly metastatic lung cancer cells. Data are representative of three independent experiments with similar results. Student’s *t* test (two-tailed, unpaired) was performed. Data represent means ± SEM. ** *p* < 0.01; ns, not statistically different.

**Figure 7 cancers-14-04121-f007:**
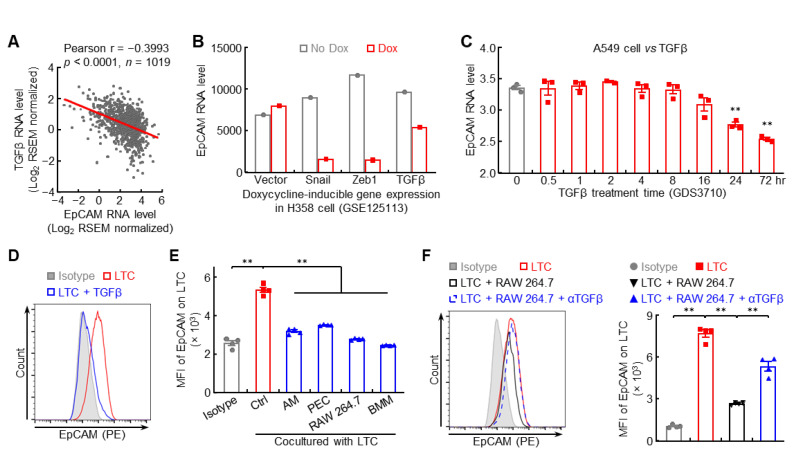
Macrophage-derived TGFβ represses EpCAM expression in lung cancer cells. (**A**) TCGA LUNG data showing negative association between EpCAM and TGFβ. (**B**) GSE125113 data showing decreased EpCAM expression in H358 cell overexpressing TGFβ. (**C**) GDS3710 data showing EpCAM downregulation by TGFβ in A549 cell (*n* = 3). (**D**) FACS data showing decreased EpCAM protein expression by TGFβ in murine lung tumor cell (*n* = 4). LTC: lung tumor cell which was established from spontaneous FVB/N lung tumor. (**E**) FACS data showing repressed EpCAM expression in murine lung tumor cells by macrophage coculture (*n* = 4). AM: alveolar macrophage, PEC: peritoneal macrophage, BMM: bone marrow derived macrophage. (**F**) FACS data showing lung tumor cell EpCAM downregulation by RAW 264.7 coculture was restored by TGFβ neutralizing antibody (*n* = 4). Data are representative of three independent experiments with similar results (**D**–**F**). Student’s *t* test (two-tailed, unpaired) was performed. Data represent means ± SEM (**C**,**E**,**F**). ** *p* < 0.01.

## Data Availability

All data are included in the article and supplemental files. The public data analyzed in the current study are available in TCGA or NCBI GEO as indicated in Methods and Figure legends. Additional data are available upon request.

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
