# Peer review of "Dynamic Expression of EpCAM in Primary and Metastatic Lung Cancer Is Controlled by Both Genetic and Epigenetic Mechanisms"

_cancers, 2022, doi:10.3390/cancers14174121_

Round 1
Reviewer 1 Report
1. How many independent experimental results were be collected for statistical analysis have to be described in all figures.
2. Previous many studies demonstrated EpCAM related-to EMT, invasion and metastasis. Today, authors show the expression levels of EpCAM are different between primary and metastatic lung cancer. Therefore, how EpCAM regulate EMT, invasion and metastasis is very important. Authors may introduce the details in the “introduction section”.
3. Many studies have demonstrated methylation and acetylation can regulate metastasis. The manuscript showed the methylation and acetylation many upregulate and downregulate the expression levels of EpCAM in primary and metastatic lung cancer respectively. Authors may discuss the possible mechanisms how methylation and acetylation regulate EpCAM expression.
Reviewer 2 Report
The article by Cui et al adequately describes the various regulatory mechanisms of EpCAM expression.
In general, the authors should consider slightly toned-down language that is used to describe EpCAM expression. For example, “These data suggest that promoter hypomethylation probably results in EpCAM upregulation” does not take into account the heterogeneity in tumor cell populations and/or the correlation between promoter status and gene expression.
In Figure 2, the authors should also include copy number variation from other adenocarcinomas. Would be interesting to discover if EpCAM gene amplification is a consistent theme.
Section 3.4. The authors should check if H827 is actually HCC827? If so, please make correction and note that A549 (KRAS mutant), H358 (KRAS mutant), will behave differently based than HCC827 (EGFR+).
A number of reports have linked EpCAM with EGFR / EMT. The authors should include the mechanistic link between the two in their conclusions.
1: Pan M, Schinke H, Luxenburger E, Kranz G, Shakhtour J, Libl D, Huang Y, Gaber
A, Pavšič M, Lenarčič B, Kitz J, Jakob M, Schwenk-Zieger S, Canis M, Hess J,
Unger K, Baumeister P, Gires O. EpCAM ectodomain EpEX is a ligand of EGFR that
counteracts EGF-mediated epithelial-mesenchymal transition through modulation of
phospho-ERK1/2 in head and neck cancers. PLoS Biol. 2018 Sep 27;16(9):e2006624.
doi: 10.1371/journal.pbio.2006624. PMID: 30261040; PMCID: PMC6177200.
2: Liang KH, Tso HC, Hung SH, Kuan II, Lai JK, Ke FY, Chuang YT, Liu IJ, Wang
YP, Chen RH, Wu HC. Extracellular domain of EpCAM enhances tumor progression
through EGFR signaling in colon cancer cells. Cancer Lett. 2018 Oct
1;433:165-175. doi: 10.1016/j.canlet.2018.06.040. Epub 2018 Jul 4. PMID:
29981429.
3: Chen HN, Liang KH, Lai JK, Lan CH, Liao MY, Hung SH, Chuang YT, Chen KC,
Tsuei WW, Wu HC. EpCAM Signaling Promotes Tumor Progression and Protein
Stability of PD-L1 through the EGFR Pathway. Cancer Res. 2020 Nov
15;80(22):5035-5050. doi: 10.1158/0008-5472.CAN-20-1264. Epub 2020 Sep 25. PMID:
32978170.
4: Hsu YT, Osmulski P, Wang Y, Huang YW, Liu L, Ruan J, Jin VX, Kirma NB,
Gaczynska ME, Huang TH. EGFR-Dependent Regulated Intramembrane Proteolysis of
EpCAM-Response. Cancer Res. 2017 Apr 1;77(7):1777. doi:
10.1158/0008-5472.CAN-16-3440. Epub 2017 Mar 22. PMID: 28330931.
5: Gires O. EGFR-Dependent Regulated Intramembrane Proteolysis of EpCAM-Letter.
Cancer Res. 2017 Apr 1;77(7):1775-1776. doi: 10.1158/0008-5472.CAN-16-2456. Epub
2017 Mar 22. PMID: 28330930.
